# Evaluation of the Impact of Catheter Ablation Procedure on Outcomes and Economic Burden in Patients with Atrial Fibrillation: Real-World Data from Italian Administrative Databases

**DOI:** 10.3390/healthcare10122561

**Published:** 2022-12-17

**Authors:** Luca Degli Esposti, Melania Dovizio, Melania Leogrande, Valentina Perrone, Roberto De Ponti

**Affiliations:** 1CliCon S.r.l. Società Benefit Health, Economics & Outcomes Research, 40137 Bologna, Italy; 2Department of Medicine and Surgery, University of Insubria, 21100 Varese, Italy

**Keywords:** atrial fibrillation, catheter ablation, healthcare-related costs

## Abstract

A real-world analysis among the Italian population has been carried out to estimate the number of atrial fibrillation (AF) patients undergoing catheter ablation and to evaluate their clinical outcome and economic burden. A retrospective analysis on administrative Italian databases has been performed. Between January 2011 and December 2019, all patients diagnosed with AF were considered and those undergoing catheter ablation were identified. Overall, 3084 (3.54%) of AF patients with at least one catheter ablation were included (mean age 63.2, 67.3% males). A significant decrease in the use of AF-related medications and in hospitalizations, mainly related to AF and heart failure, was observed during the 3-year post-ablation period. The average total cost per patient during the 1-year before ablation period was significantly higher compared to the 1-year post-ablation cost (EUR 5248 vs. 4008, respectively; *p* < 0.001). After propensity score matching, the overall mortality of patients who underwent ablation was significantly lower compared to that assessed in patients not treated with the procedure (9.386/1000 vs. 23.032/1000 person-year, respectively; *p* < 0.001). Moreover, the mean total costs were significantly higher in patients who did not undergo ablation compared to those who received ablation (EUR 5516 vs. 4008, respectively; *p* < 0.001). This real-world data analysis shows that in Italy, although catheter ablation is performed in a minority of AF patients, it is associated with significantly better post-procedure clinical outcomes and a significant reduction in healthcare-related costs.

## 1. Introduction

Atrial fibrillation (AF) is the most common sustained cardiac arrhythmia, affecting approximately 7.6 million people over 65 years in European countries (in 2016), and this number will increase by 89% to 14.4 million by 2060, with the prevalence expected to rise by 22%, from 7.8 to 9.5% [1]. A large number of patients suffer from “silent,” undiagnosed AF that often only manifests with a complication such as a stroke [2,3,4]. AF is characterized by an alteration of atrial depolarizations that result in the absence of an effective atrial contraction and a rapid chaotic rhythm, which may or may not be symptomatic.

The impact of AF on social health is significant because it is associated with high mortality and morbidity, and with significantly impaired quality of life for patients and their caregivers [5,6,7]. Patients with AF have an elevated risk of clinical complications and comorbidities. AF increases the risk for heart failure by 5.0-fold, stroke by 2.4-fold, and mortality by 2.0-fold [8]; thus, representing a critical financial burden on healthcare systems and rapidly becoming one of the world’s most significant health emergencies.

It has been reported that the annual healthcare costs for the management of AF in France, Germany, Italy, and the UK range from EUR 660–3286 million, accounting for 0.28–2.60% of total healthcare expenditures in these European countries [9]. The high cost of AF is attributable primarily to hospitalizations and complications such as stroke [9]. 

Long-term management of AF is based on the control of symptoms by either rate or rhythm control strategies in addition to prevention of thromboembolism by the use of anticoagulants [10,11]. 

Catheter ablation with pulmonary vein isolation is currently the main option for rhythm control in selected patients with paroxysmal symptomatic AF [10,12], and anticoagulation therapy is required post-ablation. The recent ESC guidelines stated that catheter ablation is recommended as second-line therapy after failure (or intolerance) to class I or III antiarrhythmic drugs [12]. Catheter ablation can be effective in the long term for maintaining sinus rhythm and improving symptoms in most patients and preventing the recurrence of AF and progression towards permanent form [13,14]. As an interventional procedure, catheter ablation of AF, as drug therapy, may be associated with complications/adverse events: in an analysis of clinical trials performed from 2005 to 2016 [15], the percentage of complications associated with catheter ablation varied from 1 to 17%, depending on the patient selection and methodologies used, while adverse events associated with drug therapy were observed in 1.4 to 23% of the patients.

A recent retrospective analysis performed among the German population between 2010 and 2017 reported that of 161,502 patients with AF or atrial flutter, 21,744 (corresponding to 13.5% of the overall patient population) underwent a left catheter ablation procedure, with a 0.05% in-hospital mortality rate [13]. A retrospective analysis among US patients evaluated the effects of catheter ablation on drug use and healthcare costs in AF patients and showed that in patients submitted to catheter ablation, compared to those who did not, a reduced antiarrhythmics utilization was observed, with a decrease of medication costs, up to 3 years post-ablation [14]. 

The recent growth of real-world evidence (“big data”) in healthcare offers an opportunity to explore the impact of medications and/or procedures on the disease burden, by using information from daily clinical practice in a large sample population [16]. In this context, limited data on the role of catheter ablation procedures among AF patients have been collected in a real-world Italian setting. Thus, this present analysis of real-world data aimed to estimate the number of AF patients undergoing the catheter ablation procedure in an Italian setting of clinical practice and evaluate their clinical outcome. Moreover, intra-patient and inter-patient assessments of healthcare costs covered by the Italian National Health System (INHS) were performed.

## 2. Materials and Methods

### 2.1. Data Source

This is a retrospective observational analysis on data extracted from secondary use from the administrative databases referred to a sample of Italian Healthcare Departments covering almost 6.4 million health-assisted individuals by the INHS, corresponding to approximately 11% of the entire national population [17,18]. The study population was representative for demographic variables of the Italian population, with a mean age of 45.2 years and 49.2% males (based in ISTAT data referred to 2017, the entire Italian population was aged on average 44.9 years, with 48.6% males [19]). Data were extracted from the following databases: (i) the demographic database, containing patient demographic data (i.e., gender, age, death); (ii) the pharmaceuticals database, containing data on drugs reimbursed by the INHS and identified by the Anatomical Therapeutic Chemical (ATC) code, number of packages, units per package, unit cost per package, and date of prescription; (iii) the hospitalization database, comprising all hospitalizations data, such as the discharge diagnosis codes classified by the International Classification of Diseases, Ninth Revision, Clinical Modification (ICD-9-CM), Diagnosis Related Group (DRG) and DRG related charge (defined by the INHS), as primary or secondary diagnosis; (iv) the outpatient specialist services database, incorporating all data about visits, diagnostic/laboratory tests for included patients (date and type of prescription, description activity and laboratory test/specialist visit charge); and (v) the payment exemption database, containing information of the exemption codes offering the contribution charge to patients for services/treatments related to a specific diseases. For this current study, Italian Entities databases were selected by their geographical distribution (by North/Centre/South of Italy), by data completeness, and by the high-quality linked datasets. 

An anonymous unique numeric code was assigned to each individual analysed to guarantee patients’ privacy (in conformity with the European General Data Protection Regulation (GDPR, 2016/679)). This unique code allowed the electronic linkage among all databases. The data were produced as aggregated form and never attributable to a single institution, or department, or doctor, or individual prescribing behaviours. The analysis has been submitted and approved by the local ethics committees of the healthcare departments involved in the analysis.

### 2.2. Study Design, Study Population and Cohort Definition

Between January 2011 and December 2019 (inclusion period), among the general population, all patients diagnosed with AF were identified and those undergoing at least one AF catheter ablation represent the group of ablated patients. The diagnosis of AF has been identified throughout all of the available period by the presence of a hospitalization discharge diagnosis (both as the main or secondary diagnosis) with the ICD-9-CM 427.31 code, while the catheter ablation procedure was identified during the inclusion period by the presence of the procedural code 37.34 (in both the primary and secondary intervention positions). The index date was defined as the date of the catheter ablation procedure, and patients were characterized during all available periods before the index date and were followed-up during all available periods after the index date (Figure 1). Only patients with at least 12 months of both characterization and follow-up period were included in the analysis; therefore, excluding patients with a time interval of <12 months between the index date and the date of the beginning or end of the inclusion period.

### 2.3. Analysis of Demographic and Clinical Patient Characteristics

At the index date, the characteristics of the patients were evaluated in terms of mean age (and standard deviation, SD), and gender expressed as the proportion of male subjects. During the characterization period, the mean time between the diagnosis of AF and the index date and the presence of previous comorbidities were assessed: tumours (identified through diagnosis of hospital discharge with the codes ICD-9-CM 140–209 and/or by the prescription of drugs with ATC code L01); chronic obstructive pulmonary disease (COPD, identified by the presence of at least one prescription of drug with ATC code R03); diabetes mellitus (identified by the presence of at least one prescription of antidiabetic drug, ATC code A10); hypertension (identified by the presence of at least one prescription for antihypertensive drugs ATC codes C02, C03, C07, C08, C09); chronic kidney disease (CKD, identified through hospital discharge diagnosis with ICD-9-CM 585 codes).

To evaluate clinical outcomes (Figure 1A), the treatments prescribed for AF and hospitalizations for AF and other cardiovascular (CV) causes were evaluated during the 12 months before and the 3 years after the catheter ablation procedure. In particular, the presence of at least one prescription of the following was assessed: antiarrhythmics (ATC code C01); beta blockers (ATC code C07); calcium channel blockers (ATC code C08); anticoagulants (ATC code B01); and, specifically, vitamin K antagonists (ATC code B01AA, the presence of the INR test was evaluated by the procedure code 907.54 among these patients), novel oral anticoagulant drugs (NOACs, Rivaroxaban, ATC code B01AF01, Apixaban, ATC code B01AF02, Edoxaban, ATC code B01AF03, Dabigatran ATC code B01AE07), heparins (ATC code B01AB)]; antiplatelet agents (ATC code B01AC) direct thrombin inhibitors (ATC code B01AE), and other antithrombotic (ATC code B01AX). In addition, hospitalizations for the following conditions were evaluated: for AF, for coronary artery disease [at least one hospitalization for acute myocardial infarction (code ICD-9-CM 410), acute cardiac ischaemia (code ICD-9-CM 411), angina pectoris (code ICD-9-CM 413), chronic cardiac ischaemia (code ICD-9-CM 414)], for cerebrovascular disease [at least one hospitalization with diagnosis of subarachnoid haemorrhage (code ICD-9-CM 430), intracerebral haemorrhage (codes ICD-9-CM 431–432), ischemic stroke (codes ICD-9-CM 434; 436), transient ischemic attack (code ICD-9-CM 435), other cerebrovascular diseases (codes ICD-9-CM 433; 437 -438)], for heart failure (code ICD-9-CM 428), for peripheral vascular disease [at least one hospitalization with diagnosis of atherosclerosis (code ICD-9-CM 440), other peripheral vascular diseases (code ICD-9- CM 443)], and for percutaneous coronary angioplasty (PTCA) [at least one hospitalization for percutaneous transluminal coronary angioplasty (ICD-9-CM code V4582) or at least one hospitalization with a percutaneous transluminal coronary angioplasty procedure (ICD-9-CM code 0066), or other coronary artery obstruction removal (ICD-9-CM 3609 code)]. 

### 2.4. Propensity Score Matching Analysis

A comparison in terms of outcomes (i.e., mortality and healthcare cost estimation) was performed in patients undergoing a catheter ablation procedure and in those without that procedure. The two cohorts were matched by applying the propensity score matching analysis (PSM, see below) to balance possible confounding variables among the two subgroups. In particular, the following covariates were considered for matching: age, gender, cancer diagnosis, COPD, diabetes mellitus, hypertension, CKD, coronary artery disease (acute myocardial infarction, acute cardiac ischemia, angina pectoris, chronic cardiac ischemia), cerebrovascular disease, heart failure, peripheral vascular disease, PTCA, and the use of antiarrhythmics, beta blockers, calcium channel blockers, anticoagulants. 

### 2.5. Analysis of Outcomes and Direct Healthcare Costs Covered by the INHS

For the intra-patient healthcare costs analysis (Figure 1B), costs were estimated in AF patients undergoing catheter ablation during the 3-year period before and after the procedure (excluding the index-date). This analysis was performed both in the overall population undergoing ablation and in the population with complete data available for the whole characterization and follow-up period of 3 years. Total average direct costs covered by the INHS, relating to pharmaceutical prescriptions (evaluated for those drugs re-imbursed by the Italian NHS, and using the INHS purchase price), hospitalizations (determined using the DRG tariffs, which represent the INHS reimbursement levels received by healthcare providers), and outpatient specialist service costs accordingly to regional tariffs, were assessed. The cost of catheter ablation procedure hospitalization was also estimated by considering regional DRG tariffs accordingly to the regional distribution of Italian healthcare departments. 

Moreover, for the inter-patient cost and outcome analysis (Figure 1C), after PSM balancing evaluation of direct healthcare costs and mortality, reported as rate per 1000 person per year (person/year), was performed in patients undergoing catheter ablation compared to those without the procedure during the first year after catheter ablation or AF diagnosis for the cohort without the procedure. The difference in costs between the two matched groups was evaluated using a generalized linear model (GLM), due to heteroscedasticity in the error variance of the cost data [20]. The GLM model was adjusted for age, gender, previous comorbidities [cancer, COPD, diabetes mellitus, CKD, hypertension, CV events (coronary artery disease, cerebrovascular disease, heart failure, peripheral vascular disease, PTCA)], and the use of antiarrhythmics, beta blockers, calcium channel blockers, and anticoagulants). Predicted costs were calculated using the coefficients from the regression analyses.

### 2.6. Statistical Analysis

Continuous variables were reported as mean and standard deviation (SD), while categorical variables were expressed as frequencies and percentages. In the cost and outcomes analyses, the results were compared, and the statistical significance was accepted for *p* values < 0.05. Mortality rates were compared by chi-square test and significance was accepted for *p* values < 0.05. The PSM was applied to compare the costs in patients diagnosed with AF in the presence and absence of ablation procedures: the two cohorts of patients were then paired (ratio 1:6 cohort with/without ablation) for the confounding variables. The standardized mean difference (SMD) was used to compare the balance of the variables between the two cohorts; Cohen et al. suggested that SMD values above 0.2 be considered small, SMD values above 0.5 considered medium-sized, and SMD values above 0.8 considered large [21,22]. In addition, a GLM model was developed to evaluate (among post-PSM cohorts) the correlation between the presence/absence of ablation procedure and healthcare costs, checking for confounding factors such as age, sex, comorbidities, treatments, and previous cardiovascular events (COPD, diabetes mellitus, CKD, tumours, hypertension, prescriptions of anticoagulants, and CV events). According to “Opinion 05/2014 on Anonymization Techniques” drafted by the “European Commission Article 29 Working Party”, the analyses involving fewer than 3 patients were not reported, as they were potentially traceable to single individuals. Therefore, results referring to ≤3 patients were reported as NI (not issuable). 

## 3. Results

### 3.1. Characteristics of AF Patients Undergoing Catheter Ablation Procedure at Baseline and during the First Three Years of Follow-Up

Among a sample of approximately 6.4 million health-assisted individuals, 110,175 patients with a diagnosis of AF were identified, corresponding to 1.71%; 86,914 AF patients were included in the analysis since they presented at least a 1-year period of data availability before and after the inclusion. Among them, 3084 AF patients (corresponding to 3.54% of overall AF patients) presented at least one catheter ablation procedure during the inclusion period. In Table 1, the demographic and clinical characteristics of AF patients undergoing a catheter ablation procedure are reported. The age of patients at the first catheter ablation procedure averaged 63.2 ± 12.4 years (67.3% were males): in particular, 22.3% of patients were under 55 years, 46.6% between 55 and 69 years, and 31.1% were over or equal to 70 years old. The time between AF diagnosis and the catheter ablation procedure averaged 7.6 ± 13.4 months. Regarding the comorbidity profile, during the characterization period, 85.3% of patients were diagnosed with hypertension, 23.1% with COPD, 14.3% with diabetes mellitus, 3.4% with CKD, and 1.3% with cancer.

The AF-related medications and CV hospitalizations of included patients during the year before and up to 3 years after catheter ablation are reported in Table 2. A significantly decrease of the proportion of patients treated with AF treatments during the post-ablation period compared to the pre-ablation period was observed. In particular, antiarrhythmics were prescribed to 73.6% of patients 1-year before, and 57.8%, 53.2%, and 51.1% during 1-year, 2-years, and 3-years after the catheter ablation procedure, respectively (*p* < 0.001). The same scenario was observed for beta-blockers, calcium antagonists, and anticoagulants. Among this class, an inverse trend was seen for NOACs use: 39.8% of patients prescribed during 1-year before and 50.2%, 43.1%, and 46.2% during 1-year, 2-years, and 3-years after the catheter ablation procedure, respectively, differently from vitamin K antagonists, used in 33.6% of patients before the procedure and in 26.1% after 1-year post-ablation, decreasing to 17.2% after 3 years (*p* < 0.001). 

From the year before up to 3 years after the catheter ablation procedure, the hospitalization rate significantly decreased. In particular, the frequency of AF-related hospitalization was 43.7% during 1-year before and 19.6%, 10.2%, and 7.4% during 1-year, 2-years, and 3-years after the catheter ablation procedure, respectively. The same scenario was observed for all the other CV-related hospitalizations. A difference of 3.4 days of hospitalization was found between the average length of hospitalization during the characterization period (9.3 days) and the first year of observation (5.9 days). 

### 3.2. Intra-Patient Analysis of Direct Healthcare Costs

As shown in Figure 2A, the mean total cost/patient, related to drugs, hospitalizations, and specialist services, was estimated during the 3 years before and after catheter ablation in all included patients. Figure 2B shows the same data for the subgroup of 1250 patients with complete data available for the whole period of 3 years.

The average total cost per patient during 1-year before ablation (EUR 5248) was significantly higher compared to the 1-year post-ablation cost (EUR 4008) (*p* < 0.001) (Figure 2A); this increase was mainly related to hospitalizations expenditure (EUR 3683 vs. EUR 2059, *p* < 0.001). In these patients, the cost related to hospitalization for catheter ablation procedure averaged EUR 5116 ± 3357, corresponding to an average daily cost of EUR 1245 ± 561, in line with the previous Italian data [23]. 

The same trend was observed in a sub-cohort of 1250 patients with 3 years of data availability before and after the catheter ablation procedure (Figure 1B). The average total cost per patient during 1-year before ablation (EUR 4709) was significantly higher compared to the 1-year post-ablation (EUR 4083) (*p* = 0.034); this increase was mainly related to hospitalizations expenditure (EUR 3311 vs. EUR 2171, *p* < 0.001) (Figure 2B). In these patients, the cost related to hospitalization for catheter ablation procedure averaged EUR 4811 ± 2878, corresponding to an average daily cost of EUR 1222 ± 545, in line with the previous Italian data [23]. 

### 3.3. Inter-Patient Analysis of Clinical Outcome and Direct Healthcare Costs

After PSM analysis of the two cohorts of AF patients, with and without catheter ablation (Appendix A), 3067 and 18402 AF patients with and without catheter ablation, respectively, were identified and included. The two cohorts were balanced for all covariates, except for age with a small difference in SMD = 0.333 and a non-clinically significant difference in the mean value (63.1 ± 12.4 vs. 67.3 ± 12.9 years) in patients with and without ablation, respectively. 

During the first year of follow-up, the mortality rate of patients who underwent ablation (9.386/1000 person-year, 95%CI: 7.425–11.347) was significantly lower (*p* < 0.001) respect to that assessed in patients not treated with the procedure (23.032/1000 person-year, 95%CI: 21.800–24.264). 

As reported in Figure 3, the mean total costs were significantly (*p* < 0.001) higher in patients without ablation (EUR 5516) than in those with ablation (EUR 4008). Analysing the single items, it emerged that the costs related to AF-related hospitalizations (EUR 1336 vs. EUR 855, *p* < 0.001), those for no AF-related hospitalizations (EUR 1680 vs. EUR 1204, *p* < 0.001), and specialist services (EUR 861 vs. EUR 506, *p* = 0.003) were significantly higher in patients without compared to those with ablation. During the first year of follow-up, 19.6% (*N* = 604) of patients with ablation respect to 23.6% (*N* = 4346) of patients without the procedure underwent AF-related hospitalization (*p* < 0.001).

After PSM and regression analyses, in patients who received a catheter ablation procedure, an annual cost reduction during the first year of follow-up per patient of EUR 709.6 was estimated (*p* < 0.001). In contrast, as shown in Table 3, the presence of comorbidities, such as tumours, diabetes mellitus, CKD, hypertension, CV events, and anticoagulant prescriptions were associated with a significant increase of costs for the management of AF patients.

## 4. Discussion

This administrative claims-based observational study provided insights into the impact of catheter ablation on disease burden in AF patients in a real-world Italian setting. In this present analysis, hospitalized patients with AF diagnosis represent almost 1.7% of the sample population. These data are in line with the literature, which reports that in Italy, AF prevalence has been estimated to be 1.85% [24]. About 3000 AF patients who underwent a catheter ablation procedure have been included; the baseline characteristics analysis showed that most patients were males (nearly 70%) and over 55 years. In the considered time frame, catheter ablation of AF was performed in a large minority of patients accounting for 3.54% of the whole population with AF. This percentage is comparable with the one of an Italian multicentre observational study performed in 2016, which reported that left atrial ablation for AF be performed in 3.8% of patients hospitalized for this arrhythmia [25]. This percentage of AF patients treated by catheter ablation is definitely lower than the one reported by König and colleagues in a study performed between 2010 and 2017 on an administrative German database, where left atrial catheter ablation accounted for 13.5% of the whole patient population hospitalized for AF [13]. Interestingly, in-hospital mortality was double in the former compared to the latter study (1.2 vs. 0.6%) [13]. Of note, the data from the present study suggest that a certain proportion of the PSM population of 18402 AF patients who did not receive ablation could include potential candidates for this procedure having a comparable demographic and clinical profile of the ablated patients. Conversely, if the two populations with and without ablation are pooled together, only 14.3% of patients have been treated by ablation. This should be also considered in light of the 1-year follow-up data showing a significantly poorer outcome in term of mortality, a significantly higher percentage of AF-related hospitalizations, and significantly higher healthcare direct costs in the PSM population without ablation. Thus, the evidence of this present analysis, for which the objective was to report data observed in the clinical practice, showed that ablation procedure is associated with a reduced mortality. This was also observed in a recent analysis among Korean patients [26], which reported a 58% reduction of the risk of all-cause death in ablated patients compared to patients receiving medical therapy [hazard ratio (HR) 0.42, 95% CI 0.27–0.65, *p* < 0.001]. Moreover, a recent meta-analysis confirmed the reduction of mortality risk (HR 0.62, 95% CI 0.54–0.72), stroke and hospitalization among patients undergoing ablation procedure compared to those treated with drug treatments alone [27]. 

The analysis of therapeutic pathways in AF patients undergoing ablation revealed a significant reduction of AF medication prescriptions, especially antiarrhythmics, and AF-related hospitalizations after catheter ablation. These data are in line with findings from a retrospective analysis among US patients, which reported that catheter ablation for AF reduced antiarrhythmic drug utilization and total prescription drug expenditures sustainably up to 3 years post-ablation [16]. Moreover, similar to what was observed in our study, a trend in the reduction of vitamin K antagonists in the post-ablation period paralleled by an increase in NOAC use has been already observed and suggests a therapeutic shifting in AF patients over time [28]. 

A significant reduction in healthcare total costs covered by the INHS was observed during the first year after the catheter ablation procedure compared to the 1 year before the procedure, up to 3 years post-procedure. The most impacting item in the cost restraining after the ablation procedure was related to hospitalization expenditure. Our data could be explained by the fact that almost 70% of included patients were hospitalized for AF at the index date or during the year before the ablation procedure, and the time between AF diagnosis and ablation procedure averaged almost 8 months. The fact that the diagnosis of AF occurred within 1 year before the ablation procedure in the majority of patients could contribute to high cost observed during this time horizon. 

Large evidence has shown that catheter ablation in patients with AF resulted in significant reductions in healthcare utilization and cost: using a large US administered database, it was found that catheter ablation, through 3 years of follow-up, restrained health care consumption mainly for AF-related admissions [14,29]. Among covariate-balanced cohorts, a significant reduction in healthcare costs was observed in AF patients who underwent catheter ablation versus those who did not, and this effect was mainly driven by hospitalization-related expenditures, both AF-related hospitalizations, and those for all other causes. These results suggested that in patients with an adequate response to medication treatments and who underwent AF related re-hospitalization, the ablation procedure is associated to healthcare cost reduction. Moreover, the generalized linear model regression analysis showed that performing a catheter ablation procedure among our population of AF patients could be associated with a cost savings of almost 700 EUR on annual patient expenditures, while the presence of comorbidities and the prescription of anticoagulants could be associated with an increase of costs. 

The limitations of this present analysis are related to its observational/retrospective nature and the results must be interpreted based on data extracted from administrative claims. The evidence of real clinical practice is derived from the evaluation of data from a sub-set of health-assisted individuals. Moreover, among administrative database limited clinical information on comorbidities and other potential confounders could be extracted, thus limiting the evaluation of their impact on the present results. Since the comorbidities analysed were addressed based on available data before inclusion (using a proxy of diagnosis, such as the use of disease-specific medications and/or disease-related hospitalizations and/or disease-specific exemption codes), there might be incomplete capture of these variables among patients. Another limitation could be related to the definition of propensity score-stratified cohorts: the cohorts with and without ablation were balanced for all variables tested except age. This difference could have influenced outcomes, such as the increase in mortality rate observed among the cohort without ablation, but it did not impact the potential eligibility for the ablation procedure among patients without the procedure. Moreover, in the definition of the PSM model, some factors not capturable from the administrative databases were not included; thus, the impact of these unknown/unmeasured confounders was not considered. The lack of a priori randomization in observational analyses generates data not powered as those which could be derived from randomized trials. Moreover, primary care data are not retrievable from administrative database.

## 5. Conclusions

Considering that very limited data on the role of catheter ablation procedures among AF patients have been collected in a clinical practice Italian setting, this real-world data analysis evaluated treatment of AF patients with catheter ablation in Italy and the impact of this procedure on outcomes and economic burden. The results showed that despite catheter ablation being used in a minority of AF patients, it is associated with a significant reduction in the use of AF medication and in hospitalizations mainly related to AF and heart failure in the 3 years post-ablation. Moreover, AF catheter ablation is associated with a significant decrease in the total healthcare expenditure, mainly related to hospitalization costs in the first year after ablation. Finally, in patients who underwent ablation compared to those of a PSM-matched population (except for the age) who did not receive the procedure, a significant reduction in all-cause mortality and healthcare costs covered by the INHS was observed.

## Figures and Tables

**Figure 1 healthcare-10-02561-f001:**
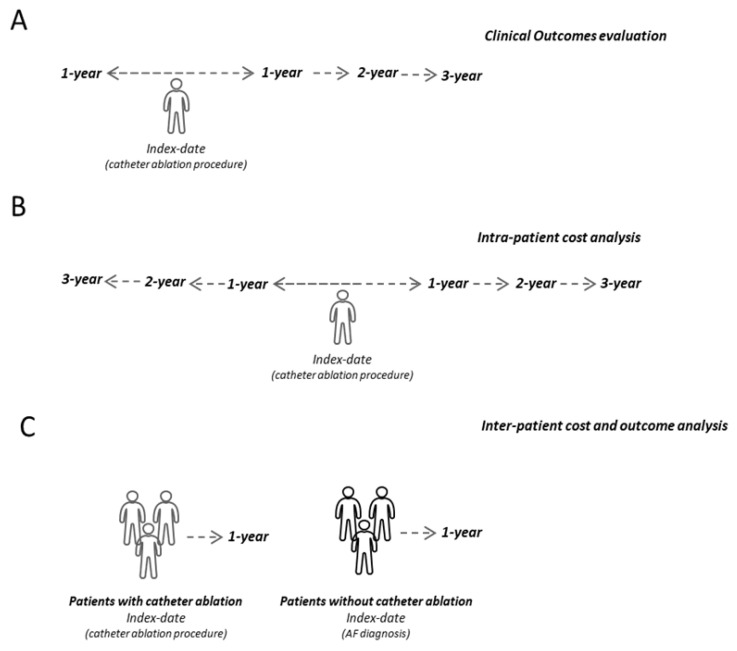
Schematic representation of the study design for clinical outcome evaluation (**A**), intra-patient cost analysis (**B**) and inter-patient cost and outcome analysis (**C**).

**Figure 2 healthcare-10-02561-f002:**
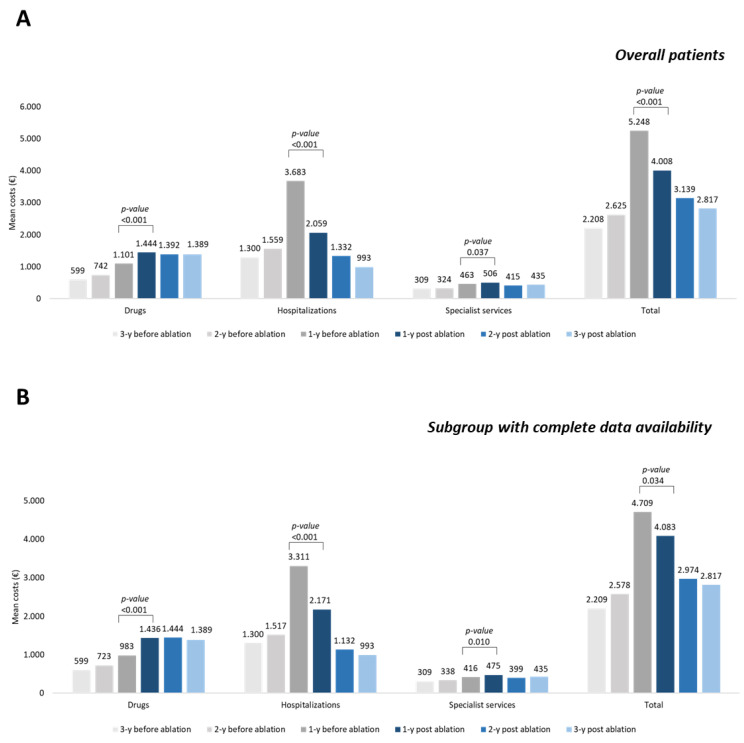
Mean annual healthcare direct costs in AF patients before and after ablation procedure, overall (**A**) and in a subgroup with complete data availability during 3 years before and after the ablation procedure (**B**).

**Figure 3 healthcare-10-02561-f003:**
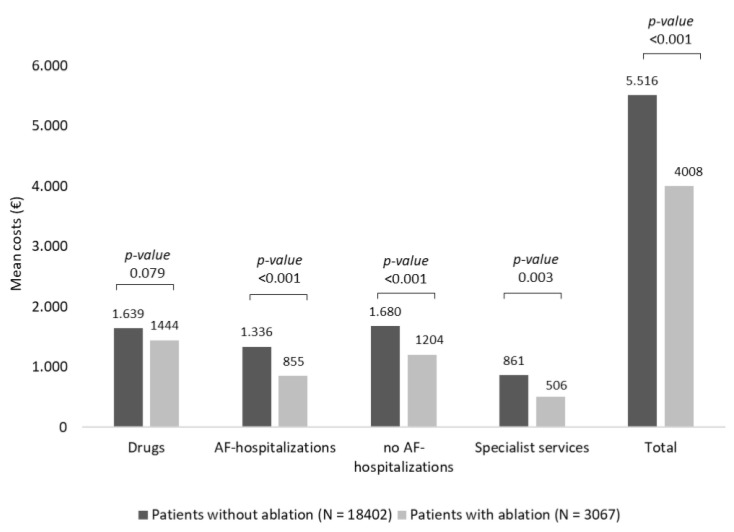
Mean annual healthcare direct costs in AF patients with and without ablation procedure, post PSM.

**Table 1 healthcare-10-02561-t001:** Baseline characteristics of AF patients undergoing catheter ablation.

	Baseline
** *Patients, n* **	** *3084* **
Age (years) at ablation procedure, mean (SD)	63.2 (12.4)
Time (months) between AF diagnosis and ablation procedure, mean (SD)	7.6 (13.4)
Age < 55 years, *n* (%)	689 (22.3%)
Age 55–69 years, *n* (%)	1437 (46.6%)
Age ≥ 70 years, *n* (%)	958 (31.1%)
Males, *n* (%)	2074 (67.3)
** *Previous comorbidities* **	
Cancer, *n* (%)	40 (1.3)
COPD, *n* (%)	711 (23.1)
Diabetes mellitus, *n* (%)	440 (14.3)
CKD, *n* (%)	105 (3.4)
Hypertension, *n* (%)	2630 (85.3)

AF, atrial fibrillation; CKD, chronic kidney disease; COPD, chronic obstructive pulmonary disease; SD, standard deviation.

**Table 2 healthcare-10-02561-t002:** Use of AF-related medications and CV hospitalizations in included patients at baseline and after catheter ablation.

	1-Year Before Ablation	1-Year After Ablation	2-Years After Ablation	3-Years After Ablation
Patients, *n*	3084	3067	2028	1250
***AF-related medications, n* (%)**				
Antiarrhythmics	2270 (73.6)	1773 (57.8) **	1079 (53.2) **	639 (51.1) **
Beta-blockers	2118 (68.7)	1849 (60.3) **	1244 (61.3) **	771 (61.7) **
Calcium channel blockers	712 (23.1)	525 (17.1) **	361 (17.8) **	228 (18.2) **
Anticoagulants	2658 (86.2)	2522 (82.2) **	1556 (76.7) **	955 (76.4) **
Vitamin K antagonists	1036 (33.6)	802 (26.1) **	429 (21.2) **	215 (17.2) **
NOACs	1228 (39.8)	1540 (50.2) **	875 (43.1) **	577 (46.2) **
*Rivaroxaban*	528 (43.0)	629 (40.8) **	326 (37.3) **	209 (36.2) **
*Apixaban*	319 (26.0)	393 (25.5) **	246 (28.1) *	170 (29.5) **
*Edoxaban*	75 (6.1)	107 (6.9) *	53 (6.1) ^§^	31 (5.4) **
*Dabigatran*	306 (24.9)	411 (26.7) **	250 (28.6) **	167 (28.9) **
Heparin	1198 (38.8)	393 (12.8) **	305 (15.0) **	156 (12.5) **
Antiplatelet agents	993 (32.2)	569 (18.6) **	435 (21.4) **	272 (21.8) **
Direct thrombin inhibitors	320 (10.4)	422 (13.8) **	257 (12.7) **	169 (13.5 ) ^@^
Other antiplatelets	37 (1.2)	12 (0.4) **	8 (0.4) ^@^	5 (0.4)
***CV hospitalizations, n* (%)**				
AF hospitalizations	1348 (43.7)	604 (19.6) **	206 (10.2) **	93 (7.4) **
Coronary artery disease hospitalizations	245 (7.9)	125 (4.1) **	89 (4.4) **	51 (4.1) **
Cerebrovascular disease hospitalizations	100 (3.2)	47 (1.5) **	44 (2.2) *	11 (0.9) **
Heart failure hospitalizations	468 (15.2)	179 (5.8) **	142 (7.0) **	54 (4.3) **
Peripheral vascular disease hospitalizations	19 (0.6)	15 (0.5)	9 (0.4)	11 (0.9)
PTCA hospitalizations	93 (3.0)	35 (1.1) **	30 (1.5) **	21 (1.7) **

AF, atrial fibrillation; CV, cardiovascular; NOACs, novel oral anticoagulant drugs; PTCA, percutaneous coronary angioplasty. Statistically significant *p* values versus the 1-year-before-ablation data were reported: ** *p* < 0.001; * *p* = 0.002; ^@^
*p* < 0.01; ^§^
*p* = 0.012.

**Table 3 healthcare-10-02561-t003:** Results of the GLM analysis of the direct healthcare costs.

Covariate	EUR	95% CI		*p*-Value
** *Catheter ablation* **	−709.6	−996.2	−423.0	<0.001
Age	28.4	19.7	37.1	<0.001
Gender	−10.7	−265.2	243.9	0.935
Cancer	6074.7	3389.2	8760.1	<0.001
COPD	890.1	515.8	1264.4	<0.001
Diabetes mellitus	1828.5	1281.1	2375.8	<0.001
CKD	7951.2	5601.6	10,300.8	<0.001
Hypertension	967.7	644.7	1290.7	<0.001
Prescriptions of anticoagulants	915.8	577.4	1254.2	<0.001
CV events	1710.1	1184.0	2236.2	<0.001
Constant	714.2	231.5	1196.9	0.004

CKD, chronic kidney disease; CV, cardiovascular; COPD, chronic obstructive pulmonary disease; GLM, generalized linear model.

## Data Availability

All data used for the current study are available upon reasonable request to CliCon S.r.l. Società Benefit, which is the body entitled to data treatment and analysis by local health units.

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
