# Peer review of "Evaluation of the Impact of Catheter Ablation Procedure on Outcomes and Economic Burden in Patients with Atrial Fibrillation: Real-World Data from Italian Administrative Databases"

_healthcare, 2022, doi:10.3390/healthcare10122561_

Round 1
Reviewer 1 Report
The authors present an analysis of real-world data showing the number of AF patients undergoing catheter ablation. The authors also present assessment of healthcare costs of patients undergoing AF ablation.
In the manuscript the real-word data are presented, what is always valuable. And interesting.
Some comments.
It is not clear how the data were extracted – was it geographical/administrative region of Italy or some random 6,4 milion population?
The introduction should be shortened – some parts can be moved to discussion. I would not mention about general strategy of AF treatment, pacemaker implantation and av nodal ablation – this has nothing to do with the topic of the study. Please consider some changes.
When writing about costs – the cost of ablation procedure was not included in the analysis? The authors noted only costs of the last year before and one year after ablation. It is interesting that in general the patients were qualified to the ablation when the costs of healthcare increased significantly (figure 2). When we compare 3-year costs before and after ablation the difference would not be probably significant.
The authors should be more cautious about differences in age between groups (line 290-292). The difference after propensity score matching is big, more than 4 years. Age is the most significant factor of total mortality. So the data should be interpreted with caution. Do we know what was the mortality rate during ablation? Does analysis include all the patients or only those who survived the ablation? It is important, because some American real-world data show that contemporary mortality rate in perioperative period of AF ablation can be as high as 1%.
The authors should include more limitations. Not only technical. Difference in age is one of them. The other is bias due to the fact that in the group of the patients not qualified to the ablation could be the patients considered, but disqualified because of some factors not captured by the study (e.g. some other comorbidities, frailty syndrome etc.)
In general discussion should be less optimistic, more critical, especially when talking about reduce in mortality.
Author Response
The authors present an analysis of real-world data showing the number of AF patients undergoing catheter ablation. The authors also present assessment of healthcare costs of patients undergoing AF ablation.
In the manuscript the real-word data are presented, what is always valuable. And interesting.
RE: We wish to thank the reviewer for the positive comment.
Some comments.
It is not clear how the data were extracted – was it geographical/administrative region of Italy or some random 6,4 milion population?
RE: Data for 6.4 million of health-assisted individuals were deriving/extracted from Local Health Units equally distributed across Italy (North/Center/South). For more clarity we added this specification in the method section, lines 109-111: “For the current study, Italian Entities database were selected by their geographical distribution (by North/Center/South of Italy), by data completeness, and by the high-quality linked datasets.”
The introduction should be shortened – some parts can be moved to discussion. I would not mention about general strategy of AF treatment, pacemaker implantation and av nodal ablation – this has nothing to do with the topic of the study. Please consider some changes.
RE: As suggested, the paragraph on the general strategy of AF treatment has been removed from the introduction. We apologize but in the introduction, due to revision requests, we added an additional paragraph reporting the complications of ablation strategy, lines 67-72: “As an interventional procedure, catheter ablation of AF, as drug therapy, may be associated with complications/adverse events: in an analysis of clinical trials performed from 2005 to 2016 [15] the percentage of complications associated with catheter ablation varies from 1 to 17%, depending on the patient selection and methodologies used, while adverse events associated with drug therapy are observed in 1.4 to 23% of the patients.”
When writing about costs – the cost of ablation procedure was not included in the analysis? The authors noted only costs of the last year before and one year after ablation. It is interesting that in general the patients were qualified to the ablation when the costs of healthcare increased significantly (figure 2). When we compare 3-year costs before and after ablation the difference would not be probably significant.
RE: We wish to thank the reviewer for the positive consideration. For the analysis of costs the index-date (hospitalization for ablation procedure) was excluded, thus we healthcare costs were estimated during the year before and after ablation. We try to explain the higher costs observed during the one year pre-ablation in the discussion, lines 375-380: “Our data could be explained by the fact that almost 70% of included patients were hospitalized for AF at the index date or during the year before the ablation procedure, and the time between AF diagnosis and ablation procedure averaged almost 8 months. The fact that the diagnosis of AF occurred within one-year before the ablation procedure in the majority of patients could contribute to high cost observed during this time horizon”.
The authors should be more cautious about differences in age between groups (line 290-292). The difference after propensity score matching is big, more than 4 years. Age is the most significant factor of total mortality. So the data should be interpreted with caution. Do we know what was the mortality rate during ablation? Does analysis include all the patients or only those who survived the ablation? It is important, because some American real-world data show that contemporary mortality rate in perioperative period of AF ablation can be as high as 1%.
RE: We wish to thank the reviewer to raise this important point. As stated by the reviewer, after PSM, the age variable showed not a perfect matching among the two cohort, with a standardized mean difference of 0.333 (thus comprising from 0.2 and 0.5; Cohen et al. suggested that SMD values above 0.2 be considered small, and SMD values above 0.5 considered medium-sized). However, the discussion/conclusion paragraph of the manuscript in accordance with the reviewer’s suggestion, being more cautionary: lines 353-362 “Thus, the evidence of the present analysis, whom objective was to report data observed in the clinical practice, showed that ablation procedure is associated with a reduced mortality. This was also observed in a recent analysis among Korean patients [26], which reported a 58% reduction of the risk of all-cause death in ablated patients compared to patients receiving medical therapy (hazard ratio [HR] 0.42, 95% CI 0.27–0.65, p < 0.001). Moreover, a recent meta-analysis confirmed the reduction of mortality risk (HR 0.62, 95% CI 0.54–0.72), stroke and hospitalization among patients undergoing ablation procedure respect to those treated with drug treatments alone [27]; lines 405-410 “Another limitation could be related to the definition of propensity score-stratified cohorts: the cohorts with ablation and without ablation were balanced for all variables tested except age. This difference could have influenced outcomes, such as the increase in mortality rate observed among the cohort without ablation, but it did not impact the potential eligibility for the ablation procedure among patients without the procedure”; lines 426-429 “Finally, in patients who underwent ablation compared to those of a PSM-matched population (except for the age) who did not receive the procedure, a significant reduction in all-cause mortality and healthcare costs covered by the INHS was observed.”
Regarding the mortality rate assessment, it was evaluated during the first year of follow-up in patients who survived the ablation; for the reviewer knowledge, the % of patients who died during the ablation hospitalization was 0.06%.
The authors should include more limitations. Not only technical. Difference in age is one of them. The other is bias due to the fact that in the group of the patients not qualified to the ablation could be the patients considered, but disqualified because of some factors not captured by the study (e.g. some other comorbidities, frailty syndrome etc.)
In general discussion should be less optimistic, more critical, especially when talking about reduce in mortality.
RE: As suggested by the reviewer, the limitation paragraph has been implemented as follow, lines 394-416: “The limitations of the present analysis are related to its observation-al/retrospective nature and the results must be interpreted based on data extracted from administrative claims. The evidence of real clinical practice is derived from the evaluation of data from a sub-set of health-assisted individuals. Moreover, among administrative database limited clinical information on comorbidities and other potential confounders could account thus limiting the evaluation of their impact on the present results. Since the comorbidities analyzed were addressed based on available data before inclusion (using a proxy of diagnosis, such as the use of disease-specific medication and/or disease-related hospitalization and/or disease-specific exemption code), there might be incomplete capture of these variables among patients. Another limitation could be related to the definition of propensity score-stratified cohorts: the cohorts with ablation and without ablation were balanced for all variables tested except age. This difference could have influenced outcomes, such as the increase in mortality rate observed among the cohort without ablation, but it did not impact the potential eligibility for the ablation procedure among patients without the procedure. Moreover, in the definition of the PSM model, some factors not capturable from the administrative databases were not included; thus, the impact of these unknown/unmeasured con-founders was not considered. The lack of a priori randomization in observational analyses generates data not powered as those which could derive from randomized trials. Primary care data are not retrievable from administrative database”
Reviewer 2 Report
1. I suggest this manuscript provide a cartoon Figure to illustrate the catheter ablation procedure.
2. Highlight the novelties of this research.
3. What’s your evidence for this conclusion that the overall mortality of patients who underwent ablation was significantly lower than that assessed in patients not treated with the procedure? Is there a randomized double-blind trial in your data?
4. Why were the mean total costs higher in patients who did not undergo ablation compared to those who received ablation?
5. What are your leading suggestions?
6. What’s the disadvantage of the catheter ablation procedure?
Author Response
- I suggest this manuscript provide a cartoon Figure to illustrate the catheter ablation procedure.
RE: We wish to thank the reviewer for this suggestion; however, the authors think that could be not properly inherent to the manuscript contents and objectives.
- Highlight the novelties of this research.
RE: The novelty of our research is related to the fact that as far from our knowledge, very limited real world data are available in Italy regarding the clinical and economic burden of AF patients undergoing ablation procedure. Thus in the conclusion paragraph, as suggested by the reviewer, we re-organized the paragraph as follow, lines 418-421: “Considering that very limited data on the role of catheter ablation procedures among AF patients have been collected in clinical practice Italian setting, this real-world data analysis evaluated treatment of AF patients with catheter ablation in Italy and the impact of this procedure on outcomes and economic burden.”
- What’s your evidence for this conclusion that the overall mortality of patients who underwent ablation was significantly lower than that assessed in patients not treated with the procedure? Is there a randomized double-blind trial in your data?
RE: We wish to thank the reviewer to raise this important point. Given the study's retrospective observational nature, the non-random assignment of patient among different cohorts can lead to not comparable groups on several characteristics based on existing data. Thus, the Propensity Score Matching (PSM) methodology was applied to abate potential unbalances in baseline characteristics among the two cohorts (AF patients with and without ablation). By definition, PSM method is a validated/accepted alternative method for the analysis of non-randomized intervention studies (such retrospective/observational studies). Regarding our research, the two cohorts were balanced for all variables assessed at baseline, with the exception for age (63.1 ± 12.4 for cohort with ablation vs. 67.3 ± 12.9 years in patients without ablation). The controlling of balancing was performed by calculating the standardized mean difference (SMD); for the variable age the SMD=0.333 (thus comprising from 0.2 and 0.5; Cohen et al. suggested that SMD values above 0.2 be considered small, and SMD values above 0.5 considered medium-sized). However, in the discussion paragraph we stated that (lines 353-361): “ Thus, the evidence of the present analysis, whom objective was to report data observed in the clinical practice, showed that ablation procedure is associated with a reduced mortality. This was also observed in a recent analysis among Korean patients [26], which reported a 58% reduction of the risk of all-cause death in ablated patients compared to patients receiving medical therapy (hazard ratio [HR] 0.42, 95% CI 0.27–0.65, p < 0.001). Moreover, a recent meta-analysis confirmed the reduction of mortality risk (HR 0.62, 95% CI 0.54–0.72), stroke and hospitalization among patients undergoing ablation procedure respect to those treated with drug treatments alone [27].” In the limitation we reported that the non at priori randomization which characterized observational analyses could generate data not powered as those generated from RTC, lines 412-413: “The lack of a priori randomization in observational analyses generates data not powered as those which could derive from randomized trials.”
- Why were the mean total costs higher in patients who did not undergo ablation compared to those who received ablation?
- What are your leading suggestions?
RE: As discussed in the manuscript, the difference in costs among the two matched cohorts was driven mainly by difference in costs related to hospitalizations and specialist services. Thus the main finding of the present analysis is that in patients with an adequate response to medication treatments who underwent AF re-hospitalization, the ablation procedure is associated to healthcare cost reduction. We reported this statement in the discussion paragraph, lines 387-389: “ These results suggested that in patients with an adequate response to medication treatments and who underwent AF related re-hospitalization, the ablation procedure is associated to healthcare cost reduction”.
- What’s the disadvantage of the catheter ablation procedure?
RE: As suggested by the reviewer, in the introduction paragraph we reported a statement on complications related to catheter ablation procedure, lines 67-72: “As an interventional procedure, catheter ablation of AF, as drug therapy, may be associated with complications/adverse events: in an analysis of clinical trials performed from 2005 to 2016 [15] the percentage of complications associated with catheter ablation varies from 1 to 17%, depending on the patient selection and methodologies used, while adverse events associated with drug therapy are observed in 1.4 to 23% of the patients.”